# Nanoscale Structural Comparison of Fibrillin-1 Microfibrils Isolated from Marfan and Non-Marfan Syndrome Human Aorta

**DOI:** 10.3390/ijms24087561

**Published:** 2023-04-20

**Authors:** Cristina M. Șulea, Zsolt Mártonfalvi, Csilla Csányi, Dóra Haluszka, Miklós Pólos, Bence Ágg, Roland Stengl, Kálmán Benke, Zoltán Szabolcs, Miklós S. Z. Kellermayer

**Affiliations:** 1Department of Biophysics and Radiation Biology, Semmelweis University, 1094 Budapest, Hungary; martonfalvi.zsolt@med.semmelweis-univ.hu (Z.M.); csanyi.csilla@med.semmelweis-univ.hu (C.C.); haluszka.dora@med.semmelweis-univ.hu (D.H.); 2Heart and Vascular Center, Semmelweis University, 1122 Budapest, Hungary; miklospolos@gmail.com (M.P.); agg.bence@med.semmelweis-univ.hu (B.Á.); rolandstengl01@gmail.com (R.S.); kalman.benke@gmail.com (K.B.); sziv.szabolcs@gmail.com (Z.S.); 3Hungarian Marfan Foundation, 1122 Budapest, Hungary; 4Department of Pharmacology and Pharmacotherapy, Semmelweis University, 1089 Budapest, Hungary; 5Department of Cardiac Surgery, University Hospital Halle (Saale), 06120 Halle (Saale), Germany

**Keywords:** fibrillin microfibrils, human, aorta, Marfan syndrome, atomic force microscopy

## Abstract

Fibrillin-1 microfibrils are essential elements of the extracellular matrix serving as a scaffold for the deposition of elastin and endowing connective tissues with tensile strength and elasticity. Mutations in the fibrillin-1 gene (FBN1) are linked to Marfan syndrome (MFS), a systemic connective tissue disorder that, besides other heterogeneous symptoms, usually manifests in life-threatening aortic complications. The aortic involvement may be explained by a dysregulation of microfibrillar function and, conceivably, alterations in the microfibrils’ supramolecular structure. Here, we present a nanoscale structural characterization of fibrillin-1 microfibrils isolated from two human aortic samples with different FBN1 gene mutations by using atomic force microscopy, and their comparison with microfibrillar assemblies purified from four non-MFS human aortic samples. Fibrillin-1 microfibrils displayed a characteristic “beads-on-a-string” appearance. The microfibrillar assemblies were investigated for bead geometry (height, length, and width), interbead region height, and periodicity. MFS fibrillin-1 microfibrils had a slightly higher mean bead height, but the bead length and width, as well as the interbead height, were significantly smaller in the MFS group. The mean periodicity varied around 50–52 nm among samples. The data suggest an overall thinner and presumably more frail structure for the MFS fibrillin-1 microfibrils, which may play a role in the development of MFS-related aortic symptomatology.

## 1. Introduction

Fibrillin-1 is one of the three fibrillin isoforms that, together and along with four latent TGF-β-binding proteins (LTBP-1, -2, -3, -4), make up the human TGF-β-binding protein-like (TB) domain protein superfamily [1]. The structure of fibrillin-1 (Figure 1) is dominated by 47 epidermal growth factor-like (EGF) domains, out of which 43 are capable of binding calcium (cbEGF) [2,3]. This property of fibrillins was found to play a role in the structural stabilization of the molecule [4,5] and protection against proteolytic degradation [6]. The EGF/cbEGF backbone of the fibrillin-1 molecule is interrupted by 7 TB domains, the fourth of which mediates interactions with integrins through the arginine-glycine-aspartic acid (RGD) cell binding site [7]. By binding the latent complexes that result from TGF-β binding to LTBPs, fibrillin-1 sequestrates TGF-β within the extracellular matrix (ECM), which is critical for tissue homeostasis and remodeling [8]. Therefore, an important role of fibrillin-1 concerns cell–matrix signaling via the regulation of TGF-β bioavailability [9]. Still, despite the substantial evidence on the growth factor regulatory role of fibrillin microfibrils, their function as structural tensiometers cannot be neglected [10].

Fibrillin-1 molecules represent the main constituents of adult fibrillin microfibrils [11]. Fibrillin-1 microfibrils are extensible ECM components that are found virtually in all adult connective tissues where they fulfil multiple tissue-specific structural and regulatory physiological functions [12]. In highly dynamic mammalian organs, such as the lung, skin, muscle, ligaments, or aortic wall, they provide an initial scaffold for the deposition of tropoelastin during elastogenesis and are retained in intimate association with the amorphous element of elastic fibers [13], contributing to the long-range elasticity of developed tissues [14]. Besides their biomechanical role, fibrillin-1 microfibrils provide tensile strength to elastin-devoid structures such as the ciliary zonule, cornea, kidney glomerulus, or tendon. Their structure is complemented by numerous microfibril-associated molecules [15,16,17] that provide an important contribution to their structure and function [8]. 

The pathophysiological importance of fibrillin-1 is highlighted by the linkage of heterozygous mutations in the fibrillin-1 (FBN1) gene (positioned on the long arm of chromosome 15) to a large family of connective tissue disorders known as type 1 fibrillinopathies [18]. The most common of these is Marfan syndrome (MFS), an autosomal dominant condition occurring with an estimated prevalence of 1 in 5–10,000 individuals regardless of sex, race, or ethnic background. More than 3000 pathogenic variants of the FBN1 gene have been linked to MFS [19]. While the disorder presents with a heterogeneous expression of clinical features, major symptoms develop in the cardiovascular, skeletal, and ocular systems. Although vascular manifestations of MFS include increased arterial tortuosity [20,21,22] and aortic aneurysms of various segments of the aorta and the main pulmonary artery [23], the main factor leading to increased mortality among MFS individuals is an aortic wall dissection that occurs in MFS individuals at significantly younger ages compared to the average population [24]. Therefore, MFS patients may benefit from prophylactic aortic surgical measures [25,26], leading to better short- and long-term outcomes by reducing disease- and surgery-related mortality rates and improving quality of life. These improvements may be achieved by establishing an effective risk stratification system centered on specific biomarkers, clinical features, and genotype–phenotype correlations [27]. 

Multiple correlations between types of mutations in the FBN1 gene and the severity or frequency of the occurrence of aortic events in MFS have been described [28,29,30,31]. However, the exact effect of the mutations on microfibrillar morphology and mechanics is yet unknown. In the present work, with the use of atomic force microscopy (AFM), we carried out a nanoscale structural characterization of MFS human aorta fibrillin-1 microfibrils associated with two different mutations in the FBN1 gene and a comparison with non-MFS microfibrillar structures. We hypothesized that isolated MFS fibrillin-1 microfibrils exhibit ultrastructural variations compared to non-MFS microfibrils, thus hinting at a potential structural role of mutant fibrillin-1 microfibrils in the MFS-specific aortic manifestations.

## 2. Results

### 2.1. Population Overview and Genetic Background

Table 1 summarizes the general characteristics of the studied populations. At the time of surgery, the MFS population (30 ± 11 years) was approximately half the age of the non-MFS group (59 ± 15 years). MFS patients were being kept under systematic observation within the Marfan outpatient clinic that operates at the Semmelweis University Heart and Vascular Center, Budapest, Hungary. Therefore, they were referred for prophylactic aortic surgery when the aortic diameter approached the threshold of 50 mm [32]. The positive diagnosis of MFS had been made previously based on the revised Ghent nosology [23]. While a family history of MFS was present in one case, ectopia lentis was not present in any of the two patients. Both subjects exhibited sufficient physical manifestations for a positive diagnosis (systemic score ≥ 7), and the presence of aortic involvement was well established. Genetic testing performed by means of next-generation and Sanger sequencing [33] identified different pathogenic mutations in the FBN1 gene (Figure 1). MFS patient #1 carried a nonsense mutation leading to the premature termination of protein synthesis at the fourth cysteine residue of the fourth cbEGF domain. In the case of MFS patient #2, a splice-site mutation was detected, likely interfering with the translational process at the level of the 27/28 cbEGF domain of the mutant fibrillin-1 molecule. 

### 2.2. Fibrillin Microfibril Bead and Interbead Ultrastructure

Regardless of disease state, the observed fibrillin microfibrils were of highly variable length, from several connected beads to more than 100 beads corresponding to lengths as large as 6–7 μm (Figure 2). The results of the investigations regarding the beads and interbeads for the groups, as well as for each sample, are presented in Figure 3. The topographical characterization of the beads revealed heights of 6.68 ± 1.15 nm and 6.13 ± 0.92 nm in the MFS and non-MFS group, respectively. Overall, mean bead height was slightly higher in the MFS group (*p* < 0.0001) (Figure 3a). Mean bead height was relatively conserved between MFS samples (*p* > 0.05), but variable within the non-MFS group, thus generating both intra- and inter-group individual differences (Figure 3b). 

Fibrillin microfibrils isolated from MFS aortic tissue had a mean bead length and width of 18.13 ± 2.89 nm and 18.56 ± 3.00 nm, respectively. By comparison, the same parameters were 25.74 ± 3.12 nm and 25.78 ± 2.91 nm, respectively, for the non-MFS group (Figure 3c,e). The inter-group differences were significant regarding both variables (*p* < 0.0001). Regarding both mean bead length and width, there were significant differences between each of the two MFS samples on the one hand and each of the non-MFS samples on the other hand (*p* < 0.001). However, there were no differences whatsoever within the two groups (Figure 3d,f).

The interbead regions appeared in the AFM images either as homogenous bands connecting the neighboring beads or split into two to three connecting arms. Occasionally, lateral arms radiating from the beads could also be observed, having a similar aspect to the ones comprising the interbead regions (Figure 2). The mean height of the interbead regions was 0.84 ± 0.40 nm for the MFS group while the non-MFS samples had a significantly higher (*p* < 0.0001) mean interbead height of 1.08 ± 0.56 nm (Figure 3g). There were no differences within the MFS group (*p* > 0.05). However, non-MFS mean interbead height values varied between individual samples, hence generating both intra- and inter-group differences (Figure 3h).

### 2.3. Fibrillin Microfibril Periodicity

Mean periodicity values measured on 15 fibrillin microfibrils of different lengths for each of the six samples were 50.90 ± 6.14 nm and 51.82 ± 6.85 nm in the MFS and non-MFS group, respectively (*p* = 0.0007) (Figure 4a). The individual mean periodicity values and their comparisons are presented in Table 2 and Figure 4b, respectively. The periodicity distribution followed similar trends between the two groups. MFS periodicity peaked at 50–52 nm (13.93%) while non-MFS periodicity drew a wider curve, peaking around 50–52 nm (12.50%) and 54–56 nm (12.24%) (Figure 4c).

## 3. Discussion

Fibrillin-1 microfibrils are key components of the aortic ECM, as their association with elastic fibers directly involves them in vascular elasticity. In MFS, the mutations in the FBN1 gene functionally compromise these microfibrillar assemblies, leading to a molecular pathomechanism responsible for the weakening of the aortic wall, thus explaining the life-threatening aortic complications that characterize the disorder [34]. Although a dysregulated fibrillin-1-dependent TGF-β signaling pathway is incriminated, mainly as the cause for MFS pathogenesis [35], an overall structural alteration of fibrillin-1 microfibrils may also contribute to MFS-specific symptomatology. The implications of FBN1 gene mutations on MFS fibrillin-1 microfibril microstructure have not yet been elucidated. Thus, investigating the structure and elasticity of MFS fibrillin-1 microfibrils is well warranted. Here, we have employed atomic force microscopy (AFM) to morphologically characterize these vital ECM elements in the human aorta. A detailed study of fibrillin-1 microfibril morphology may add information to the phenotypical characterization of individual FBN1 gene mutations, thus providing valuable evidence for the identification of concrete predictors of aortic involvement in MFS.

Based on their effect on the encoded protein, FBN1 gene mutations classify as haploinsufficient (leading to a quantitative deficiency in fibrillin-1) or dominant negative (causing abnormal protein structure) variants [36]. The nonsense (#1MFS) and splice-site (#2MFS) mutations presented in our paper both fall under the first category, thus leading to connective tissues containing mostly or only normal fibrillin-1 in reduced amounts. In accordance with the observations made by Franken et al. [37], the aforementioned mutations were associated with an increased skeletal and cardiovascular involvement but no presence of ectopia lentis. Therefore, the structure or abundance of mutated fibrillin-1 molecules may play a role in the development and severity of MFS-specific features. 

The AFM measurements revealed a slightly but significantly increased mean bead height in the MFS group. By contrast, the mean bead length and width were significantly lower in the MFS group than in the non-MFS group. In both studied directions in the horizontal (XY) plane, MFS fibrillin-1 microfibrils were characterized by shorter and thinner beads than those from all four non-MFS samples. These observations could be the result of at least two factors. On the one hand, the function of fibrillin-1 microfibrils is mediated by microfibril-associated molecules [38,39] located on the microfibrillar beads [40]. Therefore, the reduced dimensions of MFS fibrillin-1 microfibrils may be the result of a diminished quantity of associated proteins. On the other hand, a reduced amount of normal fibrillin-1, caused possibly by haploinsufficiency, may explain the more frail structure of MFS microfibril beads. Similarly, the mean interbead height was significantly smaller in the MFS group. Further studies are needed to unravel potential correlations between the effect of the FBN1 gene mutations and MFS fibrillin-1 microfibril architecture.

Individual mean periodicity values varied around approximately 50–52 nm, irrespective of pathological status. Although mean periodicity in the MFS group was found to be smaller, the difference does not seem to be conclusive, taking into account the high variability that characterized the samples in the non-MFS group. It is likely that this argument also explains the inconsistency in periodicity distribution between the two studied groups. Nevertheless, the differences do not seem to be of practical relevance, considering the magnitude of changes in periodicity that are believed to occur during physiological processes or in pathological settings. In healthy connective tissues, fibrillin-1 microfibrils have been characterized as having a specific periodicity of 50–60 nm [41,42]. Moreover, evidence points to alterations in fibrillin-1 microfibril periodicity induced by the tissue of origin [43], ionic environment [44], disease and remodeling [45], or heritable mutations in the FBN1 gene [46]. Despite periodicity being a parameter more frequently used in fibrillin microfibril studies to analyze and quantify differences in microfibrillar assemblies [47,48], the observations highlight the importance of detailed microfibrillar study through the investigation of the beads, as they make up the majority of the microfibrillar mass and are involved in a greater measure in their functionality [40,49]. 

Atomic force microscopy has proven to be a valuable tool in the morphological mapping of fibrillin-1 microfibrils of different tissular origins [43,45,48], owing to its ability to investigate biological structures in their native state. While it provides highly reliable height data (along the Z axis), the accuracy of the X and Y measurements is significantly influenced by cantilever tip geometry. As a consequence, AFM measurements of fibrillin-1 microfibrils tend to provide greater bead sizes compared to other imaging techniques, such as rotary shadowing (bead diameters of 23–29 nm) [50,51] or automated electron tomography (18–19 nm) [52]. Our findings are consistent with earlier AFM results, which reported bead width to be around 30–40 nm [41]. Therefore, it is safe to say that the MFS fibrillin-1 microfibrils assessed in our study were consistently thinner than non-MFS ones. Sherratt et al. [53] demonstrated that microfibrils act as stiff reinforcing elements within the elastic fiber. Thus, a more frail structure of the fibrillin-1 microfibrils in the MFS aorta may contribute to tissue weakening and subsequent aortic wall complications. 

It is worth noting the limitations of this study. First, considering that it includes a small number of samples, our paper does not intend to enunciate definitive conclusions on the differences between MFS and non-MFS fibrillin-1 microfibril morphology, but rather to draw attention to the possibility of concrete ultrastructural differences that also have an impact on the pathological process behind MFS-related aortic complications. As previously demonstrated, microfibrillar extension implies major conformational changes within fibrillin-1 microfibrils [52]. Considering that important elongations of more than 100 nm in periodicity imply the unraveling of the beads, MFS-related mutations affecting the fibrillin-1 domains involved in this process may lead to structural alterations in individual microfibril organization, which may be suggestive of altered packing arrangements that compromise the biomechanical properties of these microfibrillar assemblies. This affirmation needs to be supported by further studies focused on microfibrillar mechanics and elastic properties. 

Of equal importance is the establishment of TGF-β bioavailability as a predictor of MFS-related pathogenesis [54,55,56] in the context of fibrillin-1 microfibril morphological variability. Further investigations are needed to certify fibrillin microfibril structure as a reliable indicator of aortic involvement severity in MFS. Second, the fact that the non-MFS samples were obtained from subjects with different pathologies may account for the variability of values within the non-MFS group. Further analyses, including a homogeneous control group, may elucidate this aspect. Adversely, it is worth noting the homogeneity within the MFS group regarding the studied variables, which may be suggestive of consistencies characteristic of fibrillin-1 microfibril structure resulting from haploinsufficient mutations. FBN1 gene mutations have been shown to alter the process of fibrillin microfibril matrix assimilation by compromising any of its early steps (synthesis, secretion, or deposition) [57] or even by intervening at aggregation level, resulting in microfibril morphological disturbances [46]. A defective fibrillin-1 microfibril population linked with certain types of FBN1 gene mutations may explain the impairment of ECM function in the setting of MFS [58].

## 4. Materials and Methods

### 4.1. Aortic Tissue Samples

Specimens of human aorta were obtained from MFS (*n* = 2) and non-MFS (*n* = 4) individuals undergoing open-heart surgical interventions at the Semmelweis University Heart and Vascular Center, Budapest, Hungary between September 2021 and June 2022 (Table 1). The MFS patients underwent elective surgery for aneurysms of the aortic root and/or ascending aorta. Non-MFS aortic samples were obtained from individuals undergoing heart transplantation for ischemic (*n* = 1) or dilatative (*n* = 2) cardiomyopathy or aortic reconstruction due to aneurysm of the ascending aorta (*n* = 1).

### 4.2. Fibrillin-1 Microfibril Isolation

All reagents were purchased from Sigma-Aldrich, St. Louis, MO, USA. The outer layer of the aortic wall was carefully removed, and approximately 1 g of tissue was further used for the isolation of fibrillin microfibrils based on modifications of previously described methods [48,59]. The tissue was minced into fine pieces and homogenized in 5 mL of 0.05 M Tris-HCl buffer, pH 7.4, containing 0.4 M NaCl, 0.01 M CaCl_2_, 0.01% NaN_3_, and protease inhibitors (10 mM N-ethylmaleimide, 2 mM phenylmethylsulfonyl fluoride). It was further incubated with 1 mg/mL type 1A bacterial collagenase, and the digestion was allowed to proceed for 4 h at room temperature (22 °C) with gentle stirring. Following the addition of EDTA to terminate the process, the homogenate was centrifuged at 10,000× *g* for 30 min. The supernatant was labeled the low-salt extract while the residue was resuspended in 5 mL of 0.05 M Tris-HCl buffer, pH 7.4, containing 1 M NaCl, 10 mM EDTA, 0.01% NaN_3_, and protease inhibitors (10 mM N-ethylmaleimide, 2 mM phenylmethylsulfonyl fluoride) and left to be extracted for 60 h at 4 °C with gentle stirring. After centrifugation at 10,000× *g* for 30 min, the supernatant was designated the high-salt extract. Both extracts underwent size-exclusion chromatography at room temperature on a Sepharose CL-2B column (100 cm × 1.5 cm) in 0.05 M Tris-HCl buffer, pH 7.4, containing 0.4 M NaCl and 0.01% NaN_3_. The column eluent was collected in the form of 1.5 mL fractions.

### 4.3. Atomic Force Microscopy

The topography of fibrillin microfibrils adsorbed to a mica surface was investigated by using a Cypher atomic force microscope (Asylum Research, Oxford Instruments, Santa Barbara, CA, USA) employing AC160TS-R3 microcantilevers (Olympus Corporation, Shinjuku City, Tokyo, Japan) (nominal resonance frequency 300 ± 100 kHz, spring constant 26 N/m). Imaging was performed in air, in non-contact mode, at room temperature, and on fraction aliquots with various degrees of dilution. An amount of 20 μL of sample were added to a freshly cleaved mica surface, incubated for 5 min, then washed with distilled water and air-dried before visualization. All presented measurements were performed on high-salt-extract fractions. Data analysis was performed using the built-in tools of the AFM software environment (Igor Pro v6, WaveMetrics, Lake Oswego, OR, USA). On the obtained topographical images (height data), fibrillin microfibril beads were investigated for central bead height, bead width, and length, and interbead regions were assessed for height (Figure 5). A total of 100 beads for each of the 6 aortic samples (10 consecutive repeats from 10 arbitrarily selected microfibrils) were analyzed. For the study of periodicity, 15 fibrillin microfibrils were arbitrarily selected for each of the 6 aortic samples, regardless of length (as long as they comprised at least 10 beads) and measured entirely. Fibrillin microfibrils that had a highly twisted disposition or showed lateral connections (to collagen VI or other fibrillin microfibrils) were not included in the analysis.

### 4.4. Statistical Analysis

Statistical analysis was performed using the SPSS Statistics v17 software (IBM, Endicott, NY, USA). Unpaired t-test and Mann–Whitney test were used to compare the two groups (MFS vs. non-MFS). The comparisons between individual samples were conducted using the one-way ANOVA and the Kruskal–Wallis tests. The statistical significance level was set at an α of 0.05. Data are presented as mean ± standard deviation (SD).

## 5. Conclusions

Fibrillin-1 microfibril morphology is highly conserved between individuals, regardless of pathological status. We structurally characterized fibrillin-1 microfibrils isolated from two aortic samples with different FBN1 gene mutations. MFS aorta-derived microfibrillar assemblies may be structurally different from non-MFS ones, having an overall thinner architecture. The findings may reflect a loss of protein content (fibrillin-1 or microfibril-associated proteins) compared to non-MFS fibrillin-1 microfibrils, but detailed investigations are needed to elucidate how these variations influence ECM mechanics and correlate with the impairment of the TGF-β signaling pathways, thus explaining the role microfibrillar ultrastructure holds in the pathomechanism of MFS aortic symptomatology.

## Figures and Tables

**Figure 1 ijms-24-07561-f001:**
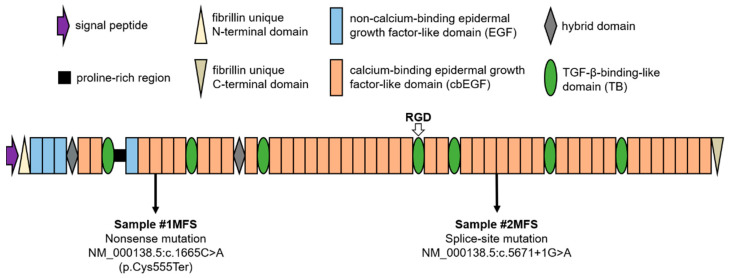
Schematic domain structure of fibrillin-1. The two fibrillin-1 gene (FBN1) mutations concerned in the current work are indicated with arrows (#1MFS, Marfan patient 1; #2MFS, Marfan patient 2). RGD, arginine-glycine-aspartic acid cell binding site.

**Figure 2 ijms-24-07561-f002:**
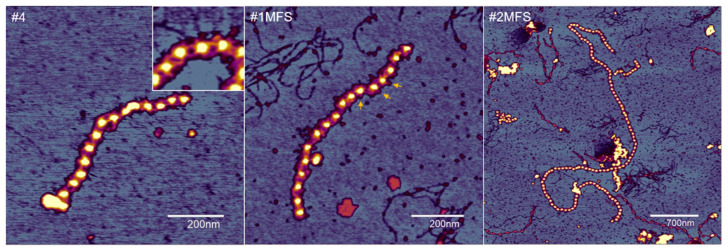
Atomic force microscopy of fibrillin microfibrils purified from MFS and non-MFS human aortic tissues; #4 is a control sample, and #1MFS and #2MFS refer to the respective mutations. The microfibrils displayed a conserved morphology but were of various lengths irrespective of the sample of origin. A characteristic “beads-on-a-string” aspect dominates the topography, in which the beads are interconnected by filamentous interbeads. As pictured in the inset, the interbeads were often visible in the form of two or three filamentous arms connecting the beads. Additionally, lateral filaments (arms) emerging from the surface of some of the beads could also be observed (yellow arrows).

**Figure 3 ijms-24-07561-f003:**
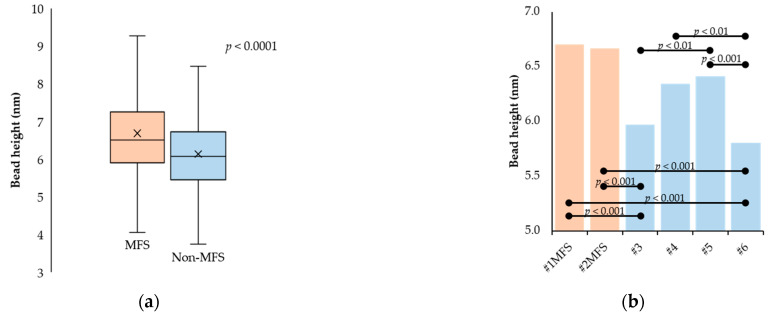
Analysis of the topographical features of fibrillin-1 microfibrils. The left side column shows the statistical comparison of the MFS and non-MFS populations in terms of bead height (**a**), length (**c**), and width (**e**), as well as interbead height (**g**). The right side column shows the statistical comparison of individual samples in terms of bead height (**b**), length (**d**), and width (**f**), as well as mean interbead height (**h**). The horizontal black bars mark the statistically significant differences between sample means (except for bead length and width, where each of the MFS samples generated statistically significant differences with each of the non-MFS samples).

**Figure 4 ijms-24-07561-f004:**
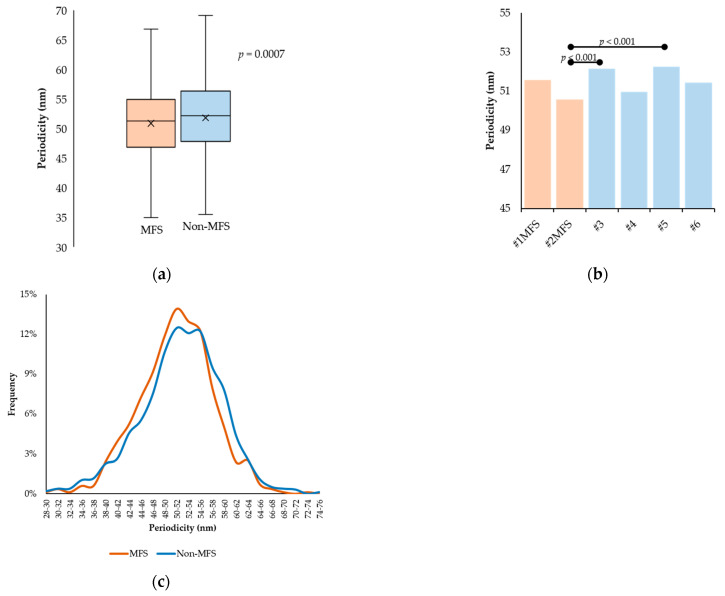
Comparison of interbead periodicity in MFS and non-MFS fibrillin microfibrils. (**a**) Statistical comparison of pooled periodicity values in the two populations. (**b**) Comparison of mean periodicities between individual samples. The horizontal lines indicate pairs of samples displaying statistical differences. (**c**) Distribution of periodicity in the two studied groups.

**Figure 5 ijms-24-07561-f005:**
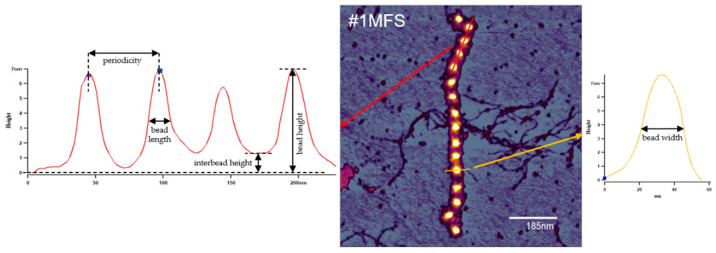
Schematic representation of the sections used for measuring the structural parameters of fibrillin-1 microfibrils. Considering the arbitrary disposition of the microfibrillar assemblies, bead length (longitudinal section, red) and width (transversal section, yellow) were assessed in respect to the median axis of the respective microfibril.

**Table 1 ijms-24-07561-t001:** Summary of the two studied populations and diagnostic characteristics of the MFS subjects.

	MFS		Non-MFS			
Sample/Patient	#1	#2	#3	#4	#5	#6
**Sex**	F	M	M	F	M	M
**Age**	37	17	66	55	34	72
**Family history**	Positive	Negative				
**Aortic involvement**	Present					
**Ectopia lentis**	Absent					
**FBN1 mutation**	Present					
**Systemic features ***	Wrist and thumb signs					
	Pectus excavatum	Pectus carinatum				
	Pes planus					
	Reduced US/LS ** + increased arm/height + no severe scoliosis	Scoliosis				
	Skin striae	-				
	Myopia ≥ 3 diopters	-				
	Mitral valve prolapse					
**Systemic score**	9	8				

* according to the revised Ghent nosology [23] ** US/LS, upper segment/lower segment ratio.

**Table 2 ijms-24-07561-t002:** Summary of the fibrillin microfibril periodicity results both per group and per individual sample.

	Periodicity			
	Number of Values Registered (n)	Mean (nm)	Minimum (nm)	Maximum (nm)
**MFS group**				
#1MFS	276	51.57 ± 6.32	31.22	72.86
#2MFS	564	50.57 ± 6.03	28.46	69.36
**Non-MFS group**				
#3	298	52.17 ± 7.29	22.69	69.69
#4	281	50.97 ± 6.27	29.59	68.85
#5	618	52.27 ± 6.68	31.71	71.37
#6	355	51.43 ± 7.11	31.74	95.54

## Data Availability

Data available on request due to ethical restrictions.

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
