# Peer review of "Nanoscale Structural Comparison of Fibrillin-1 Microfibrils Isolated from Marfan and Non-Marfan Syndrome Human Aorta"

_ijms, 2023, doi:10.3390/ijms24087561_

Round 1

Reviewer 1 Report

The paper reports interesting data on morphology of Fibrilli-1 microfibrils, obtained by atomic force microscopy, when the purified patient sample is adsorbed on mica surface. Using scanning probe technique to correlate morphological differences with mutations of MFS is quite innovative, however in my view the data is too few to assess a statistical validation of the hypothesis. It might be important to add other experimental results even if not supporting the thesis of structural frailness of the mutated fibrils captured by AFM. 

Besides a larger number of sample for MFS positives, it would be important to have a force-distance recording for the samples. The authors can also explain why it was not possible to perform the experiment in liquid setup, when protein is hydrated. 

Some aspects of the papers which can be improved are the following:

The results on the change of periodicity seem not conclusive, because the shift of the center of two distributions reported in Fig. 4c might be included in the statistical variance of the sample.

Did authors evaluate an effect on another geometric parameter (i.e. tortuosity) between MFS and non-MFS samples which can be included in the results and discussion?

Authors may better explain with a graphic what width and length represent on the analysed bead since they are extracted from  images with arbitrary axis orientation.

How periodicity was calculated? From profiles, or autocorrelation, or FFT transform?

One typo in the description of AFM methodology about the volume of solution incubated on mica.

Author Response

Thank you very much for the careful evaluation of our manuscript and the helpful and constructive suggestions! The questions and comments helped us improve our manuscript further. Please find below our point-by-point responses to the comments:

  1. Thank you for pointing out one of the major drawbacks of this study, which is the relative scarcity of the data. We do realize this shortcoming, which stems partly from the limited number of human surgical samples, from the relatively lossy protein preparation procedure, and from the somewhat poor attachment of the microfibrils to the substrate surface. In the manuscript, we briefly address this issue in the Discussion section.
  2. Indeed, analyzing the mechanical properties of the fibrillin-1 microfibrils has been one of our objectives. We have investigated the local elastic properties of the microfibrils by using a Fast Force Mapping (FFM) approach under aqueous conditions. Alas, we did not obtain conclusive results, most likely to the poor attachment of the fibrils to the substrate. We are currently working on employing alternative substrates and attachment procedures. Due to the premature nature of the FFM data we opt not to include them in the current manuscript.
  3. Thank you for drawing attention to this. Indeed, we need to make a more careful conclusion based on the pooled statistics. We addressed this issue in the Discussion section of the revised manuscript (lines 209-216).
  4. Indeed, tortuosity could be used as an additional parameter based on which further comparisons could be made between the different types of fibrillin-1 microfibrils. In our view, tortuosity is related to the more quantitative persistence length, which could provide access to the bending rigidity (hence elastic properties) of the fibrils. Our current aim is to combine this approach with direct mechanical (FFM) measurements.
  5. Bead width and length were measured with respect to the median axis of the respective microfibril. We added an additional figure in the Materials and Methods section of the manuscript (Figure 5) in which we exemplify the sections used for assessing the studied parameters.
  6. Periodicity was defined as the distance between the centers of two consecutive beads measured on the median line of the respective microfibril. A schematic representation can be seen in the newly added Figure 5.
  7. We corrected the typo in line 311 of the revised manuscript, thank you for drawing our attention to it.

Reviewer 2 Report

Marfan syndrome is a well-described autosomal dominant syndrome with widely variable clinical manifestations. In majority of cases it is caused by mutations in the gene for fibrillin-1, the main constituent of extracellular microfibrils, which consequently affects microfibrillar function and microfibrils’ supramolecular structure.  Effects of various mutations on microfibrillar morphology are highly relevant to our understanding of this disease, but are still largely unknown

In my opinion, the paper contains an original contribution to scientific knowledge and topic is suitable for publication in International Journal of Molecular Sciences. Text is well written and concise with only few, minor errors, and figures are illustrative and informative. The paper can be accepted for publication in this form, with only few modifications.

.

Specific comments:

The title clearly reflects the content. However, I would recommend to change the order of the words to emphasize the main subject. For instance: Nanoscale structural comparison of fibrillin-1 microfibrils isolated from Marfan- and non-Marfan- syndrome human aorta

Figure 2. It seems to me that all 3 compared pictures (control and two mutations) should be photographed at the same magnification. In picture order, control should be either first or last, not in the middle. Smaller square in the control picture should be mentioned and described. Please check the magnification in #2MFS.

Figure 3. The numbers on y-axis of B, G i H graphs should be written with dot, not comma

Line 144: please describe more detailly how lateral arms look. I do not see the description or indicator to lateral arms in Figure 2.

Author Response

Thank you very much for the attentive evaluation of our manuscript and the helpful and constructive suggestions! Please find below our point-by-point responses to the comments:

  1. We agree with your recommendation and have made the modification accordingly.
  2. The reason for the third panel of Figure 2 having a different magnification was to exemplify the difference between isolated microfibril segments, which were highly variable in respect to length, irrespective of the sample of origin. The other two panels with higher magnification are intended to offer a closer look at the structural details of the microfibrillar assemblies. Indeed, it would be best if the two mutated samples were pictured next to each other. Therefore, we switched the positions of the first two panels. We also added a more detailed description of the inset panel in the caption of Figure 2.
  3. We made the changes accordingly, thank you for drawing our attention to it.
  4. We marked some of the lateral arm structures in Figure 2 with yellow arrows. We modified the description of the lateral arms in lines 151-152 of the revised manuscript.

Reviewer 3 Report

The authors of this manuscript isolated and investigated the fibrillin-1 microfibirlis from Marfan syndrome aortic tissue with AFM technique. As authors acknowledged the sample numbers were limited and preliminary, the analytical technique and AFM characterization approach does present scientific merits. To improve the novelty and significance of the manuscript, it's recommended additional sample numbers to be included, and elaboration on molecular mechanisms with respect to sarcomere structure or length-tension relationship to be discussed. 

Author Response

Thank you very much for the careful evaluation of our manuscript and the positive feedback!

Thank you for pointing out one of the major drawbacks of this study, which is the relative scarcity of the data. We do realize this shortcoming, which stems partly from the limited number of human surgical samples, from the relatively lossy protein preparation procedure, and from the somewhat poor attachment of the fibrils to the substrate surface. In the manuscript, we briefly address this issue in the Discussion section.

Regarding the second recommendation, in the Discussion section of the manuscript, we acknowledged the importance of Marfan syndrome-related mutations with respect to the relationship between fibrillin-1 microfibril periodicity and bead structure. To the best of our knowledge, there are currently no such studies involving Marfan syndrome tissue. We are currently working on analyzing the mechanical properties of the fibrillin-1 microfibrils by employing a Fast Force Mapping (FFM) approach.